# Derazantinib Inhibits the Bioactivity of Keloid Fibroblasts via FGFR Signaling

**DOI:** 10.3390/biomedicines11123220

**Published:** 2023-12-05

**Authors:** Shuqia Xu, Yongkang Zhu, Peng Wang, Shaohai Qi, Bin Shu

**Affiliations:** 1Department of Plastic Surgery, The First Affiliated Hospital of Sun Yat-sen University, Guangzhou 510080, China; xushq@mail.sysu.edu.cn; 2Department of Burn Surgery, The First Affiliated Hospital of Sun Yat-sen University, Guangzhou 510080, China; zhuyk1994@126.com (Y.Z.); wangp276@mail.sysu.edu.cn (P.W.); 3Department of Burn and Plastic Surgery, Shenzhen Institute of Translational Medicine, The First Affiliated Hospital of Shenzhen University, Shenzhen 518025, China

**Keywords:** derazantinib, fibroblast growth factor receptor, keloid

## Abstract

Keloids are common benign cutaneous pathological fibrous proliferation diseases, which are difficult to cure and easily recur. Studies have shown that fibroblast growth factor receptor-1 (FGFR1) was enhanced in pathological fibrous proliferation diseases, such as cirrhosis and idiopathic pulmonary fibrosis (IPF), suggesting the FGFR1 pathway has potential for keloid treatment. Derazantinib is a selective FGFR inhibitor with antiproliferative activity in in vitro and in vivo models. The present study determined the effects of derazantinib on human keloid fibroblasts (KFs). Cell viability assay, migration assay, invasion assay, immunofluorescence staining, quantitative polymerase chain reaction, Western blot analysis, HE staining, Masson staining, and immunohistochemical analysis were used to analyze the KFs and keloid xenografts. In this study, we found that derazantinib inhibited the proliferation, migration, invasion, and collagen production of KFs in vitro. The transcription and expression of plasminogen activator inhibitor-1 (PAI-1), which is closely related to collagen deposition and tissue fibrosis, was significantly inhibited. Also, derazantinib inhibited the expression of FGFR1 and PAI-1 and reduced the weight of the implanted keloid from the xenograft mice model. These findings suggest that derazantinib may be a potent therapy for keloids via FGFR signaling.

## 1. Introduction

During the course of wound healing processes, the mechanical environment at the wound site changes spatiotemporally through the formation of fibrin clots as a temporary extracellular matrix (ECM) and the proliferation and migration of wound healing-related cells such as fibroblasts, myofibroblasts, epidermal keratinocytes, and many types of immune cells. Following skin injury, wound repair necessitates an interaction between various cell types, wherein fibroblasts have a crucial role in the later stages of the healing process. In adults, wound healing leads to the formation of a scar, characterized by nonfunctional fibrotic tissue. Keloids are common benign cutaneous pathological fibrous proliferation diseases with locally invasive characteristics and a high incidence of recurrence [1,2]. Keloids occur in susceptible individuals, usually due to injury reaching the reticular dermis and abnormal wound healing [3]. Abnormal cellular responses, such as the increased proliferation of fibroblasts and inflammatory cells and vascular disturbance, can lead to keloids [3,4]. Single-cell RNA sequencing revealed a significant increase in fibroblasts, endothelial cells, mural cells, and mast cells in keloid tissue [5]. Given their pivotal role as primary mediators of scarring and fibrosis, fibroblasts are a key focus of research aimed at enhancing wound healing. Keloid fibroblasts (KFs) have been shown to produce high levels of collagen, fibrous junctions, and elastin, promoting extracellular matrix accumulation during keloid development [6]. Mast cells have the potential to contribute to keloid formation by releasing inflammatory molecules, including histamine and cytokines, which can trigger neighboring fibroblasts [7]. Many growth factors, such as transforming growth factor β (TGF-β), fibroblast growth factor (FGF), and vascular endothelial growth factor (VEGF), play an essential role in these actions [8,9]. However, the interaction between keloid cell functions and the fibrotic cascade remains unclear, which restricts clinical interventions for keloids.

Keloids can be treated with surgery followed by radiotherapy or conservative treatment, such as intralesional corticosteroid injection, compression therapy, and intralesional 5-fluorouracil injection. Inexpensive intralesional corticosteroid injections remain the most common conservative therapy. Intralesional corticosteroid injections probably act by decreasing inflammatory cytokine production. Corticosteroids combined with 5-fluorouracil injections induce keloid regression more effectively [10]. However, the disadvantages include recurrence, injection-induced pain, systemic side effects, and local side effects of corticosteroids. New therapeutic approaches are needed.

The FGF/FGF receptor (FGFR) signaling pathway plays a crucial role in regeneration and tissue repair [11]. Disrupting astrocytic FGFRs reduced scar size without negatively impacting neuronal survival in the injured brains of mice [12]. FGFR1 expression was found to be increased in pulmonary fibrosis-related diseases [13]. A recent study also showed that the expression levels of FGFR1 were enhanced in adults suffering from cirrhosis [14]. Given the close association between FGFR1 and tissue regeneration and fibrotic disease, it is reasonable to assume that FGFR1 has clinical significance in the treatment of keloids.

Derazantinib is a selective FGFR inhibitor with potent activity against FGFR1, FGFR2, and FGFR3 kinases [15]. A multicenter phase 1–2 clinical trial showed that orally administered derazantinib was safe and effective for advanced intrahepatic cholangiocarcinoma patients [16]. Patients with advanced solid tumors received drug treatment at an initial dose of 25 mg every other day and dose escalation from 25 to 425 mg every day. Partial responses and tumor reduction were observed [17]. Furthermore, derazantinib demonstrated antiproliferative activity in in vitro and in vivo models, such as cultured chondrocytes and xenograft models with endometrial cells [18,19]. In this study, we hypothesized that derazantinib could attenuate keloid formation by directly targeting KFs. To test this hypothesis, we detected the expression of FGFR1 in keloid tissue and normal skin. We explored the inhibition mechanism of derazantinib on the bioactivity of KFs. In addition, the therapeutic potential of derazantinib was evaluated in an athymic nude mouse model.

## 2. Materials and Methods

This study was approved by the Medical Ethics Committee of the First Affiliated Hospital, Sun Yat-sen University No.2020[020]. Written informed consent was obtained from all participants. Animal experiments were reviewed and approved by the Institutional Animal Care and Use Committee of the First Affiliated Hospital, Sun Yat-sen University. *BALB/C* female athymic nude mice were procured from Guangdong Medical Laboratory Animal Center (Guangzhou, China). All animals were kept at the Laboratory Animal Center of Sun Yat-sen University. Standard animal care protocols were implemented. In addition, the mice were kept in pre-sterilized cages and placed in a laminar flow environment, with each mouse having its own cage. Sterilized mouse food and water were used.

### 2.1. Immunofluorescence Assay of Keloid Tissue and Normal Skin Tissue

Eight patients (four females and four males, aged 25 to 46 years) with untreated keloids on the anterior chest, not sun-exposed, were recruited. The nature of the keloid tissues was confirmed using clinical and histological evidence. Normal skin was harvested from six female patients (aged 27 to 42 years) who had undergone breast reduction operations. Keloid tissue and normal skin tissue were fixed in 4% paraformaldehyde and sectioned. The slides were incubated with the following primary antibodies overnight at 4 °C: FGFR-1 (60325-1-Ig, Proteintech, Rosemont, IL, USA, mouse) and collagen I (66948, CST, Danvers, MA, USA, mouse) antibodies were diluted according to the instructions. Next, the slides were washed with PBS and immersed in the corresponding goat anti-mouse (ab6789, Abcam, Waltham, MA, USA, 1:10 000). DAPI (Thermo Fisher, Bend, OR, USA) was used for nuclear counterstaining. Images of FGFR-1 (green)-, collagen I (red)-, and DAPI nuclear-stained (blue) cells were obtained using a fluorescence microscope (Olympus, Tokyo, Japan) and analyzed using ImageJ 1.53 software.

### 2.2. Isolation and In Vitro Culture of Fibroblasts from Human Keloid

Keloid samples were obtained from keloid resection procedures and stored at low temperatures. Then, the subcutaneous fat and epidermis of the keloid samples were removed using a scalpel and scissors. The remaining keloid tissues were minced and incubated in Dulbecco’s Modified Eagle Medium (DMEM, Gibco, Grand Island, NY, USA) supplemented with 0.3% collagenase IV (Biofroxx, Einhausen, Germany) and 0.2% hyaluronidase (Biofroxx, Einhausen, Germany) for two hours at 37 °C. The cells were separated with a 40 μmol filter screen and centrifugated at 1500 rpm for 8 min. After digestion, the KFs were sustained in DMEM supplemented with 10% fetal bovine serum (FBS; Gibco, USA) and 1% penicillin–streptomycin (Gibco, USA). Then, the cells were seeded onto 10-cm culture dishes and cultured in a humidified atmosphere at 37 °C with 5% CO_2_. Passages 3 through 6 cells were analyzed. Cells were randomly divided into treatment groups and control groups. The assay was repeated with three cell samples from independent donors. Participants who were responsible for outcome evaluation were not aware of the group’s allocations.

### 2.3. Chemical Reagents

Derazantinib (MCE, Monmouth Junction, NJ, USA) was dissolved in 100% dimethyl sulfoxide (DMSO) to a stock concentration of 10 mM and stored at −20 °C. The reagent was diluted directly to the desired dose upon use, with an identical final concentration of DMSO in both the experimental and control groups. The final DMSO concentration was maintained below 0.1% in all experiments to ensure there was no cytotoxic effect on the cultured cells.

### 2.4. Cell Counting Kit-8 (CCK-8) Assay of KFs

The drug effect on KFs proliferation was analyzed with CCK-8 (Dojindo Laboratories, Mashiki, Japan). The effect of the drug on the fibroblasts (Fbs) from normal skin was also analyzed with CCK-8. The cells were successively digested, centrifuged, and counted. Then, 1.0 × 10^3^ cells were seeded in each well of 96-well culture plates and treated with different concentrations of derazantinib (0, 0.16, 0.31, 0.63, 1.25, 2.5, and 5 µmol/L). The viability of the cells was tested at the indicated time points. Concisely, 100 μL culture medium and 10μL CCK-8 reagent were added per well, and the cells were incubated for 3 h at 37 °C according to the protocol. In addition, the optical density values at 450 mm of each well were measured with a microplate reader (Thermo Fisher, Waltham, MA, USA). The CCK8 assay with normal fibroblasts was repeated with three cell samples from independent donors.

### 2.5. Cell Migration Assay of KFs

For the scratch assay, 1.0 × 10^4^ KFs were seeded in each well of a 6-well culture. When the cells in the 6-well plates reached more than 90% confluence, the cell monolayer was scraped with a 200 μL pipette tip and washed with PBS three times. Then, the cells in plates were cultured with different concentrations of derazantinib (0, 0.31, 0.63, 1.25, and 2.5 μmol/L) for 48 h. Photographs were taken after the scratching at the indicated time points (0 h, 24 h, and 48 h). The migrated area was counted with ImageJ software.

### 2.6. Cell Invasion Assay of KFs

For the transwell assay, 6 × 10^4^ serum-starved KFs were placed with 2.5 μmol concentrations of derazantinib in the upper well of a Matrigel-coated membrane (24-well, 8 μm, Corning, Corning, NY, USA). After 48 h incubation, the chamber was fixed with methanol and stained with 0.1% crystal violet (Servicebio, Wuhan, China). Cell numbers in five randomly selected fields were counted using Image-Pro Plus 6.

### 2.7. Cell Viability Staining

When the KFs in the 6-well plates reached 90% confluence, the cells were treated with 2.5 μmol concentrations of derazantinib for 48 h. The Calcein AM and Propidium Iodide assay kit (Beyotime, Shanghai, China) was performed to assess the effect of derazantinib on cell apoptosis. A total of 100 µL Calcein AM working solution was added to each well and incubated at 37 °C for 20 min. Then, the Calcein AM working solution was removed, and 100 µL PI working solution was added to each well and incubated at 37 °C for 5 min. The PI working solution was removed and washed with Hank’s Balanced Salt Solution twice for 15 min. Images of the cells were obtained using a fluorescence microscope (Olympus, Tokyo, Japan).

### 2.8. Immunofluorescence Assay of KFs

First, 2 × 10^5^ KFs per well were seeded in 24-well plates at room temperature for 30 min in a regular culture medium. Second, the medium was replaced with DMEM with 2.5 μmol/L concentrations of derazantinib for 48 h and fixed in 4% paraformaldehyde overnight. The cells were incubated with the following primary antibodies: collagen I (66948, CST, USA, mouse), Ki-67 (ab16667, Abcam, USA, rabbit), PAI-1 (13801-1-AP, Proteintech, IL, USA, rabbit), and α-smooth muscle actin (α-SMA) (AF1032, Affinity, Changzhou, China, rabbit). Then, the cells were washed with PBS and immersed in the following secondary antibodies: collagen I (Anti-Mouse IgG Alexa Fluor 488, CST, USA), Ki-67 (Anti-Rabbit IgG Alexa Fluor 488, CST, USA), PAI-1 (Anti-Rabbit IgG Alexa Fluor 488, CST, USA), and α-SMA (Anti-Rabbit IgG Alexa Fluor 488, CST, USA). DAPI was used for nuclear counterstaining. Images of antibodies-positive (green), actin (red), and DAPI nuclear-stained (blue) cells were obtained under a fluorescence microscope (Olympus, Tokyo, Japan). Image-Pro Plus was used for fluorescence intensity analysis.

### 2.9. Quantitative Reverse Transcription (RT)-PCR Analysis of KFs

KFs treated with different concentrations of derazantinib (0, 1.25, 2.5, and 5 μmol/L) for 48 h were washed with PBS. Then, 1 ml Trizol reagent (Invitrogen, Carlsbad, CA, USA) was added to crack the KFs. A quantitative PCR analysis was performed using the SYBR Green PCR Kit (Servicebio, China) in a real-time thermal cycler (Bio-Rad CFX96, USA). GAPDH was used as an internal control. Furthermore, gene expression was quantified using the 2−ΔCt method, where the 2−ΔCt method: A = CT (target gene, sample) − CT (GAPDH), B = CT (target gene, control) − CT (GAPDH), K = A–B, fold-change = 2^−K^. The primers for real-time qPCR analysis were sourced from the NCBI reference sequence, as listed in Table 1.

### 2.10. Western Blot of KFs

KFs were treated with 2.5 μmol derazantinib or without derazantinib for 48 h to examine protein expression. Radioimmunoprecipitation assay (RIPA) buffer was used to collect cell lysates [20]. Cell lysates were separated with sodium dodecyl sulfate–polyacrylamide gel electrophoresis (SDS-PAGE) and transferred onto a polyvinylidene fluoride (PVDF) membrane [21]. The PVDF membrane was separately treated with the specific primary antibodies: α-SMA (AF1032, Affinity, Changzhou, China, rabbit), collagen I (66948, CST, USA, mouse), PAI-1 (13801-1-AP, Proteintech, IL, USA, rabbit), FGFR1 (60325-1-Ig, Proteintech, USA, mouse), GAPDH (GB11002, Servicebio, Wuhan, China, rabbit), PI3K (AF5112, Affinity, Changzhou, China, rabbit), p-AKT (AF0016, Affinity, Changzhou, China, rabbit), AKT (BS0115R, Bioss, Wuhan, China, rabbit), p-ERK (AF1015, Affinity, Changzhou, China, rabbit), ERK (AF0155, Affinity, Changzhou, China, rabbit), TGF-β (BA0290, Bosterbio, Beijing, China, rabbit), Smad (6367, Affinity, Changzhou, China, rabbit), and p-Smad (3367, Affinity, Changzhou, China, rabbit). Then, the washed PVDF membrane was treated with horseradish peroxidase-conjugated secondary antibody (ab205718, Abcam, USA, goat anti-rabbit IgG) for 2.5 h. Ultimately, the target protein was shown with the enhanced chemiluminescence (AI800, GE, Marlborough, MA, USA) detection system.

### 2.11. Local Injection Treatment of Keloids with Derazantinib versus Glucocorticoid in an Athymic Nude Mouse Model

Twelve *BALB/C* female athymic nude mice (20−22 g), eight weeks old, were kept at the Laboratory Animal Center of Sun Yat-sen University. Keloids from the chest were used for transplanting into three patients (aged 25 to 46 years). Keloid tissue from each patient was transplanted subcutaneously into 4 mice, respectively. Before implantation and after tissue removal, the weight of the keloid was measured with an electronic balance (accuracy of 1 mg). After removal, the keloid tissue was minced into 5 mm × 5 mm × 5 mm fragments and immediately stored in an ice box. The mice were anesthetized with phenobarbital sodium. Then, four small keloid tissue samples were implanted subcutaneously on the back of each mouse as follows: one each on the upper left back, lower left back, upper right back, and lower right back. One and two weeks after implantation, each keloid implant received 0.1 mL of a different injection: normal saline (upper left back), 2 mg/mL derazantinib (DZB) (lower left back), 1 mg/mL derazantinib (upper right back), and 2 mg/mL betamethasone (BMT) (lower right back). The keloid tissue was extracted four weeks after implantation. A check for a normal distribution of weight loss was performed before analysis. HE staining, Masson staining, and immunohistochemistry were performed on each extracted tissue sample. For HE staining, sections of keloid tissue were immersed in Harris Hematoxylin stain and EOSIN stain in sequence. Masson’s staining was performed using Masson’s kit (Servicebio, Wuhan, China). Tissue biopsies were stained according to the manufacturer’s protocols. The results of weight loss were averaged per keloid donor. For immunohistochemistry, the slides were incubated with the following primary antibodies overnight at 4 °C: FGFR-1 (60325-1-Ig, Proteintech, USA, mouse) and collagen I (66948, CST, USA, mouse) antibodies or PAI-1 (13801-1-AP, Proteintech, IL, USA, rabbit) and α-SMA (AF1032, Affinity, Changzhou, China, rabbit) antibodies were diluted according to the instructions. Next, the slides were washed with PBS and immersed in the corresponding goat anti-mouse (ab6789, Abcam, USA, 1:10,000). DAPI was used for nuclear counterstaining.

### 2.12. Statistical Analysis

All data are presented as means ± standard deviation (SD). Student’s *t*-test was used to conduct two experimental group comparisons. All analytical graphs were analyzed using GraphPad Prism 9 (La Jolla, CA, USA). * *p* < 0.05 was considered statistically significant.

## 3. Results

### 3.1. FGFR1 Demonstrated Higher Expression in Keloid Tissue

As FGFR1 has been proven to be essential for tissue regeneration and pulmonary fibrosis [22,23], we first detected FGFR1 expression in human keloid tissues. FGFR1 demonstrated higher expression in keloid tissues than in normal skin (Figure 1A).

### 3.2. Derazantinib Reduced the Viability and Promoted the Apoptosis of KFs

The CCK-8 assay demonstrated a significant inhibitory effect of derazantinib on cell proliferation in a dose-dependent manner compared with the control group. Treatment with derazantinib at 5 μmol/L for 48 h almost completely wiped off the KFs (Figure 1B). Treatment with derazantinib at 2.5 μmol/L showed a significant inhibitory effect on KFs, whereas the inhibition of Fbs was not significant at the same concentration of derazantinib. However, treatment with derazantinib at 5 μmol/L showed a significant inhibitory effect on Fbs. In addition, the Ki67 immunofluorescence staining demonstrated a significant inhibitory effect of derazantinib on KFs proliferation (Figure 1C). The Calcein AM-PI staining showed that derazantinib promoted the apoptosis of KFs (Figure 1D).

### 3.3. Derazantinib Inhibited the Migration and Invasion of KFs

The migratory and invasion capability of KFs is essential to the pathogenesis of a keloid. We found that derazantinib significantly inhibited the migration of KFs (Figure 2A,B) and also suppressed their invasion capability (Figure 2C).

### 3.4. Derazantinib Suppressed the Activation of KFs

The effects of the drugs on the expression of ECM deposition factors, including collagen I, α-SMA, and PAI-1, were investigated using immunofluorescence analysis. Immunofluorescence staining showed that all markers were robustly reduced in KFs after treatment (Figure 3B–D).

At the transcriptional level, treatment with increased concentrations of derazantinib significantly suppressed the gene expression of PAI-1 (Figure 3A), with significant differences among the groups (* *p* < 0.05). The expression of PI3K and JNK genes was also suppressed in the experiment group (Figure 3G). No significant effect was found on FGFR1 mRNA expression levels.

As demonstrated by the Western blot (Figure 3E), treatment with the drug significantly suppressed the expression of PAI-1, FGFR1, collagen I, and a-SMA. Additionally, it also suppressed the expression of ERK, p-ERK, p-AKT, and PI3K (Figure 3H), which represented the inhibition of the AKT and ERK signaling pathways. No significant effect was found on TGF-β and SMAD expression levels.

### 3.5. Derazantinib Inhibited Collagen Production and Disrupted Angiogenesis of Keloid Tissue in the Athymic Nude Mouse Model

The transplantation of human keloid tissue explanted under nude mouse skin can maintain the pathological state of human keloid cells, which is a suitable animal model for testing candidate anti-keloid drugs [24]. The weight loss of the implanted keloid tissue was averaged: normal saline (59 ± 8.3 mg), 2 mg/mL DZB (104 ± 10.6 mg), 1 mg/mL DZB (92 ± 7.1 mg), and 2 mg/mL BMT (85± 8.8 mg). A statistically significant difference was found between the treated derazantinib group and the control group (Figure 4A). Masson staining and HE staining showed less abundant collagen and micro-vessels in the derazantinib treatment groups (Figure 4B). As demonstrated by the immunofluorescence staining of the keloid tissue explants, the expression of FGFR1 and collagen I was lower in the derazantinib treatment groups than in the control group (Figure 4C). Consistently, the expression of a-SMA and PAI-1 was suppressed in the derazantinib group (Figure 4D).

## 4. Discussion

A keloid is an abnormal proliferation of scar tissue beyond the boundaries of the original wound with an overabundant accumulation of extracellular matrix. Due to the incomplete understanding of their pathogenesis, keloids have a high recurrence rate and unsatisfactory clinical management [25]. Recent single-cell RNA sequencing showed a significant expansion of KFs, suggesting that fibroblasts are closely related to the pathogenesis of keloids [26]. Aberrant amplification of fibroblasts has become a potential therapeutic target for keloids.

Derazantinib is a novel selective FGFR inhibitor with proven efficacy and manageable toxicities in treating FGFR-driven tumors and congenital chondrodysplasias [27,28]. In this study, we demonstrated that derazantinib presented a pleiotropic effect on KFs, which included reduced viability, inhibited migration and invasion, and the downregulation of the expression of PAI-1, a-SMA, and collagen I. Furthermore, in the xenograft nude model, we also demonstrated that intralesional injection of derazantinib inhibited collagen production and reduced the weight of transplanted keloid tissue. Lower FGFR1 expression may be due to cell apoptosis or necrosis. The limitation is the lack of detection of the phosphorylation status of FGFR1 and other kinases. We will complete these experiments in further research.

Research on fibrotic diseases has described significant FGFR1 expression in idiopathic pulmonary fibrosis (IPF) patients, which is associated with fibroblast migration and increased MAPK-signaling, contributing to the pathogenesis of IPF [29]. The transcription and expression of PAI-1 was significantly downregulated by derazantinib. Under pathologic conditions, excessive PAI-1 contributes to the excessive accumulation of collagen and other ECM proteins around the wound, thus preserving scarring [30]. It has been reported that PAI-1 inhibits uPA/tPA activity and significantly affects MMP-dependent remodeling, which is a crucial molecule affecting the infiltration and degradation of fibroblasts [31,32]. PAI-1 siRNA was found to inhibit alveolitis and pulmonary fibrosis in BLM-treated rats by inhibiting the proliferation and promoting apoptosis of fibroblasts via the ERK and AKT signaling pathways [33]. Meanwhile, PAI-1 is a major promoter of vascular-related pathologies [34,35]. However, in some situations, increased expression of PAI-1 suppresses cardiac fibrosis by inhibiting TGF-β and myofibroblast activation [36]. In the present study, derazantinib induced the downregulation of PAI-1 levels, which could have interfered with the accumulation of the ECM and prevented keloid formation. Also, the derazantinib treatment groups showed disrupted angiogenesis in a xenograft model. As an anti-angiogenic drug [37], derazantinib may have a detrimental effect on vascular morphogenesis in the xenograft model. However, the athymic nude mice model cannot satisfactorily re-create the in vivo condition of a keloid in its native state. Further experiments are needed to clarify the anti-proliferative mechanism of derazantinib and the molecule mediating PAI-1. In addition, immunofluorescence and Western blot assays showed that α-SMA (myofibroblast marker) and collagen I (collagen marker) expression were decreased in the derazantinib group. These data suggest that derazantinib alleviates fibrosis in keloids.

Although derazantinib at 2.5 μmol/L showed different inhibitory effects on KFs and Fbs, it inhibited the proliferation of Fbs at higher concentrations. Despite the pleiotropic effect on KFs, the toxic effects of the drug on the cells cannot be completely excluded. This evidence suggests that derazantinib is less likely to be KF-specific and that some toxic effects on normal cells are possible. This is the limitation of the drug. However, fluorouracil, an antineoplastic medication, is used in clinical settings to treat keloids with intralesional injections. When administering keloid injections, clinical trials can be conducted in the early stages to confirm the effectiveness of derazantinib. Previous research showed that the drug was well-tolerated with manageable toxicities in a non-selected patient population and demonstrated single-agent antitumor activity in patients [17]. Especially for patients with multiple incurable keloids, the safety of systemic medication makes it a potential therapeutic option.

In summary, we demonstrated that the selective FGFR inhibitor derazantinib suppressed KFs bioactivity. Derazantinib targeting keloid therapy may be a promising and effective therapeutic strategy that warrants further clinical trials.

## Figures and Tables

**Figure 1 biomedicines-11-03220-f001:**
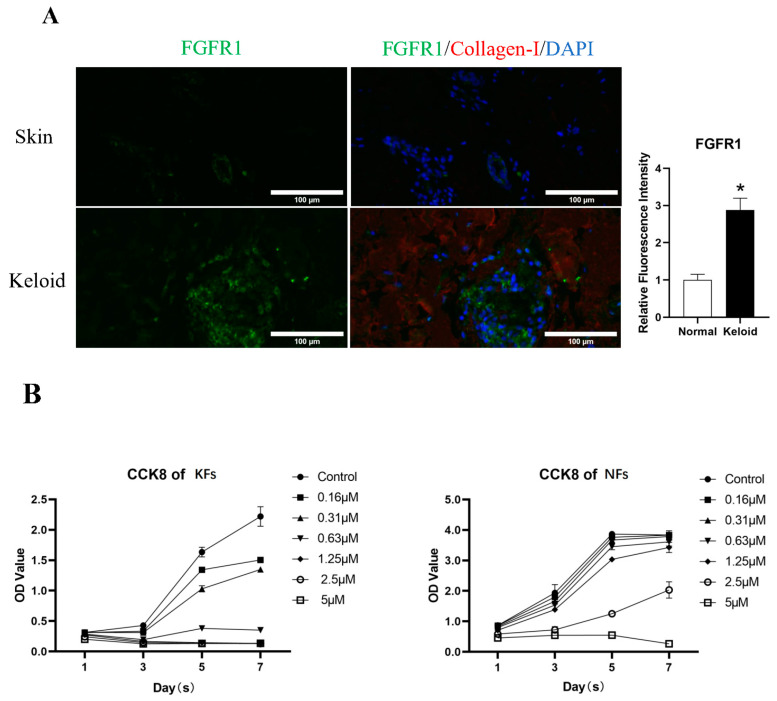
(**A**) Immunofluorescence staining of keloids and normal skin. The length of the scale is 50 μm. (**B**) The CCK-8 assay demonstrated a significant inhibitory effect of derazantinib on the proliferation of keloid fibroblasts (KFs). (**C**) Ki67 immunofluorescence staining showed that Ki67 expression was decreased in the experiment group. The length of the scale is 50 μm. (**D**) The Calcein AM-PI staining showed that Calcein-AM-positive cells were decreased and PI-positive cells were increased in the experiment group. The length of the scale is 100 μm. * *p* < 0.05.

**Figure 2 biomedicines-11-03220-f002:**
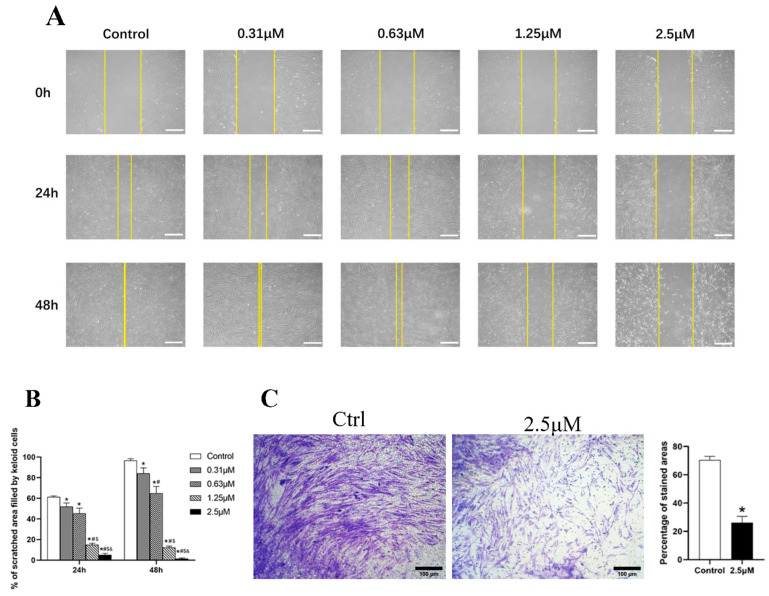
(**A**,**B**) The scratch assay showed that derazantinib inhibited the migration of KFs. The length of the scale is 500 μm. * *p* < 0.05 compared with the control group, # *p* < 0.05 compared with the 0.31 μM group, $ *p* < 0.05 compared with the 0.63 μM group, and & *p* < 0.05 compared with the 1.25 μM group. (**C**) The transwell assay showed that derazantinib inhibited the invasion of KFs. The length of the scale is 100 μm.

**Figure 3 biomedicines-11-03220-f003:**
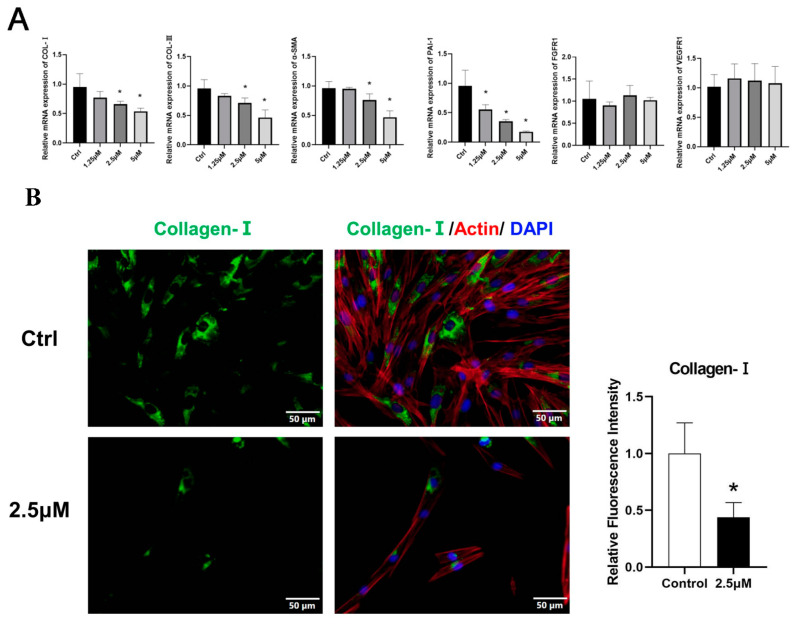
KFs were treated with 2.5 μmol derazantinib or without derazantinib for 48 h. (**A**) The PCR showed that the expression of fibrotic genes was suppressed in the experiment group. (**B**–**D**) The immunofluorescence staining showed that collagen I, α-SMA, and PAI-1 expression were decreased in the derazantinib group. The length of the scale is 50 μm. (**E**,**F**) The WB showed that the protein production of a-SMA, collagen I, PAI-1, and FGFR1 in the derazantinib group was suppressed. (**G**) The PCR showed that the expression of PI3K and JNK genes was suppressed in the experiment group. (**H**) The WB showed that the expression of ERK, p-ERK, AKT, TGF-β, p-AKT, and PI3K in the derazantinib group was suppressed. No significant effect was found on the SMAD expression levels. * *p* < 0.05 compared to control group.

**Figure 4 biomedicines-11-03220-f004:**
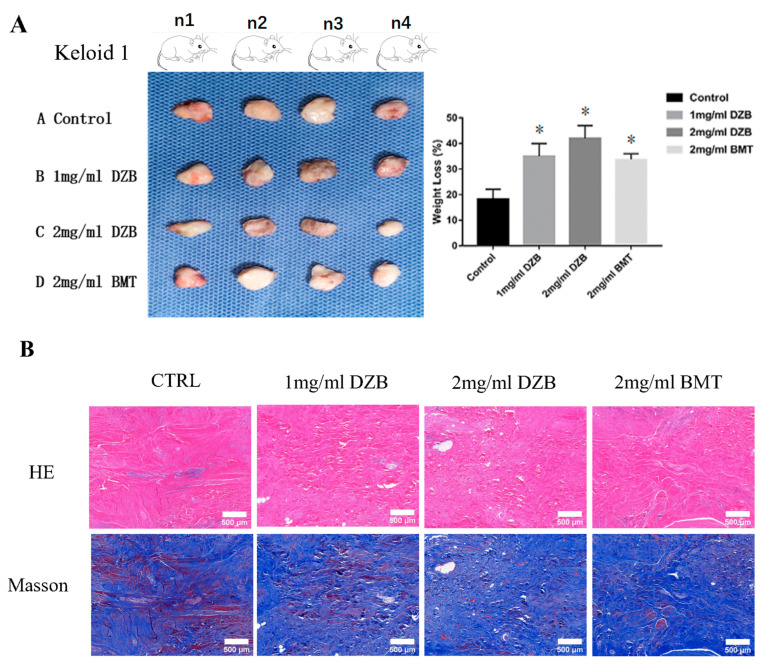
(**A**) In an athymic nude mouse model (*n* = 12), keloid tissue from each patient (*n* = 3) was transplanted subcutaneously into 4 mice, respectively. The weight loss of the keloids transplanted into the nude mice was statistically different between the control group and the derazantinib group. * *p* < 0.05 compared with the control group. (**B**) Masson staining and HE staining showed less abundant collagen and micro-vessels in the derazantinib groups. The length of the scale is 500 μm. (**C**) The immunofluorescence staining showed that the expression of FGFR1 and type I collagen was decreased in the derazantinib group compared with the control group. The length of the scale is 50 μm. (**D**) The immunofluorescence staining showed that the expression of PAI-1 and a-SMA was consistently reduced in the derazantinib group compared with the control group. The length of the scale is 50 μm. (DZB—derazantinib; BMT—betamethasone).

**Table 1 biomedicines-11-03220-t001:** Primers used in quantitative PCR analysis.

Gene	Primer Sequence	Product Size (bp)
*H-GAPDH-S*	GGAAGCTTGTCATCAATGGAAATC	168
*H-GAPDH-A*	TGATGACCCTTTTGGCTCCC	
*H-VEGFR1-S*	GCACCTTGGTTGTGGCTGA	155
*H-VEGFR1-A*	CTCTCCTTCCGTCGGCATT	
*H-FGFR1-S*	GAGGCTACAAGGTCCGTTATGC	292
*H-FGFR1-A*	CCAATCTTGCTCCCATTCACCT	
*H-CTGF-S*	GCCCAGACCCAACTATGATTAGAG	207
*H-CTGF-A*	GGATGCACTTTTTGCCCTTCT	
*H-PAI-1-S*	CCCCACTTCTTCAGGCTGTT	189
*H-PAI-1-A*	GCCGTTGAAGTAGAGGGCAT	
*H-COL1-S*	CCCCTGGAAAGAATGGAGATG	104
*H-COL1-A*	AGCTGTTCCGGGCAATCCT	
*H-COL3-S*	CCCCGTATTATGGAGATGAACC	109
*H-COL3-A*	CCATCAGGACTAATGAGGCTTTC	
*H-α-SMA-S*	CAATGTCCTATCAGGGGGCAC	209
*H-α-SMA-A*	CGGCTTCATCGTATTCCTGTT	
*H-PI3K-S*	TACACTGTCCTGTGCTGGCTA	295
*H-PI3K-A*	GAGATTCCCATGCCGTCGTA	
*H-TGFβ1-S*	GGAGAAGAACTGCTGCGTGC	132
*H-TGFβ1-A*	TCCAGGCTCCAAATGTAGGG	
*H-SMAD2-S*	TGCCACGGTAGAAATGACAAG	230
*H-SMAD2-A*	TAACAGACTGAGCCAGAAGAGC	
*H-SMAD3-S*	CTACCAGTTGACCCGAATGTGC	74
*H-SMAD3-A*	TCTGTCTCCTGTACTCCGCTCC	
*H-JNK-S*	TCTCCAACACCCGTACATCAA	151
*H-JNK-A*	CTCCTCCAAGTCCATAACTTCCT	

## Data Availability

Data are available upon request.

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
