# Peer review of "Derazantinib Inhibits the Bioactivity of Keloid Fibroblasts via FGFR Signaling"

_biomedicines, 2023, doi:10.3390/biomedicines11123220_

Round 1

Reviewer 1 Report

Comments and Suggestions for Authors

The manuscript entitled "Derazantinib inhibits bioactivity of keloid fibroblasts via FGFR signaling" presents in vitro and in vivo results supporting the use of the inhibitor ARQ087 (derazantinib) against the development of keloids. Although the scientific question is interesting, the manuscript suffers at the level of experimental design.

Especially regarding the in vitro experiments, the authors have shown in Figure 1 that inhibitor concentrations from 2.5 to 5 μM provoke apoptosis and death of the keloid fibroblasts (KFs). However, they have not used the inhibitor in normal fibroblast cultures. If the inhibitor provokes also death of normal fibroblasts at the same concentration range, then its clinical significance is limited.

Furthermore, the authors used the same higher ARQ087 concentrations (2.5 and 5 μM) in many of the experiments presented in Figure 3, in order to show the implication of FGFR1 and other downstream genes in the effects of the inhibitor on KFs. However, using concentrations leading to apoptosis is not indicated for mechanistic studies, since many of the observed changes may be due to cell death. Beyond the concentration selection, the authors do not state clearly the time-point at which these experiments were performed. Based on the information of Materials and Methods, mRNA analysis was performed after 48 hrs of treatment (page 3, line 132), immunofluorescence at the same time-point (page 3, line 123), but Western analysis was performed only after 2 hrs (page 4, line 143). So, it is difficult for me to understand how after 48 hrs of ARQ087 treatment the mRNA of FGFR1 remains unaltered (Figure 3A), while the protein levels of FGFR1 seem to be drastically down-regulated already after 2 hrs (Figure 3E-3F).

Moreover, the study of the inhibitor effects should focus primarily on the phosphorylation status of FGFR1 (instead of its protein levels), since this is a kinase inhibitor. Also, since in the paper by Hall et al. (ref. 21 in the manuscript), the IC50s against other kinases (RET, FMS, DDR2, PDGFRβ) are similar to the one against FGFR1, the authors should also study these kinases, in order to justify the title of the manuscript. Especially PDGFRβ could also mediate the same effects as FGFR1, in terms of viability, migration, and downstream targets such as ERK and AKT.

Other comments:

FGFR1 and collagen expression are not visible in Figure 1A

Why is only calcein AM (and not PI) quantitation shown in Figure 1D?

The authors should check carefully the manuscript for inadvertent errors, such as the inclusion of lines 245-248 in page 11 (first paragraph of Discussion).

Comments on the Quality of English Language

There are many language errors. I am quoting some of them:

page 1, line 33 should read "vascular disturbance" instead of "disturbance vascular "

page 1, line 35 should read "Many" instead of "Much"

page 1, line 44 should read "expression levels of FGFR1"

page 2, line 72 should read "fixed" instead of "followed by fixation"

page 11, lines 259-260 should read "down-regulation of the expression" instead of "down-regulating the expressions"

Reviewer 2 Report

Comments and Suggestions for Authors

As currently presented this work is not reproducible. Key methodological details are not reported. The data is overinterpreted and sometimes misinterpreted. That said, the work may contribute a useful addition for the clinical management of keloids. This is highly pertinent to the authors’ local community, which experience a high incidence of keloids. 

The authors are not native speakers of English; the grammar is poor. 

For these reasons I recommend the authors be offered Resubmission and review after MAJOR REVISION.

Ln 58. Please provide evidence substantiating “This study was approved by the Medical Ethics Committee of the First Affiliated Hospital, Sun Yat-sen University”; and, also for “Institutional Animal Care and Use Committee of Sun Yat-sen University.” i.e. Authorisation Ref#.

Ln 72. Please identify the source and clone ID for “primary and secondary antibodies”? Please identify the origin and diluents for “...according to the instructions”? More detail is necessary to permit reproducibility.

Ln 74. Please provide detail. How were “Images… obtained under a fluorescence microscope”?

Ln 78. Please provide detail. How were “keloid samples… obtained through an operation” (sic)? For example; tissue, location, sun-exposure/not-sun exposed, articulating/non-articulating, etc?

Ln 79. Please provide detail. How were “aseptic scissors… used to remove the epidermis…”? The skill required to practice this technique is exceptional!

Ln 81. Please provide detail. What is the diluent for “0.3% collagenase IV…  and 0.2% hyaluronidase”? More detail is necessary to permit reproducibility.

Ln 84. What is the evidence isolated cells are “…separated KFs”? What vessels were used to “sustain KFs… in DMEM…”?

Ln 88. What are “independent cell samples”? Are these representing different donors, different cultures (same donor), different sample tissue harvests (same donor, different time)? In other words, are these technical replicates, biological replicates, or clinical replicates?

Ln 94. It is not clear to what substance has “final concentration was maintained below 0.1%”?

Ln 97. Please identify the abbreviation “CCK-8”? Is this a substance, a technique, a product number?

Ln 98. Please reformat “1.0×103 cell”? This is meaningless.

Ln 99. It is stated that cells were “treated with different concentration (sic) of derazantinib (0, 0.16, 0.31, 0.63, 1.25, 2.5, and 5 μmol) for 48 h.” However, measures were recorded “after 24, 48, 72, and 96 hours”? I am confused. How do I reproduce this experiment?

Ln 103. What is an “enzyme-labeled instrument”?

Ln 108. Please explain what is meant by “vigorously scraped with a 200μl pipette tip”? How was this technique verified to be reproducible?

Ln 112. “transwell” should Transwell™. This is a registered Trade Name.

Ln 113. For reproducibility, more technical detail is required for “Boyden chamber coated with Matrigel”. For example: membrane material? pore size? concentration of Matrigel? (another Trade Name!), depth of Matrigel?

Ln 114. For reproducibility, how were “invaded cells… fixed and stained”? 

Ln 115. What is the origin of “randomly selected fields”? Were these images? How were they captured?

Ln 117. How was “90% confluence” determined? Is this a guess, and estimate? How was it measured?

Ln 118. For reproducibility, how was “Calcein AM and Propidium Iodide staining… performed”?

Ln 122. For reproducibility, what is “a regular culture medium”?

Ln 127. For reproducibility, please identify the “Antibodies against collagen I, Ki-67, PAI-1, 127 and α-smooth muscle actin (α-SMA)”? Are these monoclonal or polyclonal? What is the animal of original? Who is the supplier? Please specify the band-pass filters used for fluorescence imaging?

Ln 132. For reproducibility, how were “KFs… collected”?

Ln 134. For reproducibility, please identify the “reverse transcription kit”?

Ln 137. Please revise “RNA expression degrees”. I am not familiar with this measure!

Ln 138. For reproducibility, please identify the origins (design and source) of “primers for real-time qPCR analysis”? I presume that “primers” (laboratory slang) means oligonucleotides? Please clarify.

Ln 142. Please revise the sentence “For protein production in cells, KFs were treated with 2.5μmol derazantinib or with-142 out derazantinib for 2 h to examine the cellular signaling molecules.” This is grammatically incorrect. This sentence has two subjects: “protein production” and “cellular signaling molecules”. These are not equivalent. It could be argued that they are mutually exclusive! This reader is confused!

Ln 145. For reproducibility, please identify “RIPA lysis buffer”?

Ln 145. For reproducibility, how were “Equal amounts of proteins” determined?

Ln 147. What is the origin of the “immunoblot”? This has not been previously identified.

Ln 148. For reproducibility, please identify “blocking solution (5% skim milk)”? What is the original source of “milk”?

Ln 149. For reproducibility, how were “polyvinylidene fluoride… treated by horseradish peroxidase-conjugated secondary antibody”?

Ln 151. For reproducibility, how is “the target protein… shown by the enhanced chemiluminescence”?

For common techniques, I strongly recommend referring to the original citation for the method.

Ln 156. For reproducibility, please identify the origin of “the mice”? As a minimum the strain/genotype should be identified.

Ln 157. For reproducibility, please specify how “tissue samples were implanted on the back of… mice”? The use of adverb “on” suggests tissues were simply laid onto the dorsal surface. How were tissues retained? How were animals prevented from removing tissues? Was fur removed? How were tissues prevented from dehydrating?

Substantially greater experimental detail is required.

It is later reported (Ln 225) that “human keloid tissue explanted under nude mouse skin can maintain the pathological state of human keloid cells.” Does this imply that, rather than “on the back”, in these experiments human tissues were implanted sub-cutaneously, beneath the dorsal skin? Please clarify! Moreover, what is the evidence that in these reported experiments, these observations by others is verified?

Ln 159. For reproducibility, please specify how “each implant received 0.1ml of a different injection”?

Ln 160. Throughout this report dosage is reported in µM; however, injected doses are reported in the Materials and Methods as 2mg/ml (sic) derazantinib, 1mg/ml (sic) derazantinib (upper right back), and 2mg/ml (sic) betamethasone. This inconsistency is unhelpful for others attempting to reproduce this experiment.

Ln 162. For reproducibility, please specify how “tissue was extracted four weeks after implantation”?

Ln 173. What is the basis for the claim “FGFR1 demonstrated higher expression in keloid tissue than in normal skin”? How was this expression measured? I find Figs 1A and C, very difficult to interpret due to weak signal, poor contrast and low resolution. The scale bars are illegible! I am not convinced by the reported data.

Ln 176. No evidence is presented to support the claim “CCK-8 assay demonstrated a significant inhibitory effect of derazantinib on KFs proliferation in a dose dependent manner.” Data is presented from only a single dose!

Ln 177. Fig 1D does not report scale bars, as claimed in the legend.

Ln 180. I would argue that “Calcein AM-PI staining showed that derazantinib promoted apoptosis of KFs” is overinterpretation. A single source of data is insufficient to conclude death is due to apoptosis. The data reported in Fig 1B could be interpreted to indicate derazantinib is cytotoxic at doses >2.5 µM. This is also evident in Fig 2A. What is the evidence apoptosis is the mechanism?

Ln 192. The claim that “derazantinib significantly inhibited the migration of KFs in a dose-dependent manner” based upon evidence reported in Fig 2A, is tenuous. The data illustrated in Fig 2C indicates cytotoxicity; it is overinterpretation to conclude “derazantinib inhibited the invasion of KFs”. It is also not made clear, which side of the Boyden chamber is illustrated in Fig 2C. Scale bars are not reported in Fig 2C.

Ln 203. My observation of Fig 3B-D is identical to Fig 2; resolution, illumination, contrast, and size conspire to ensure it is difficult to interpret these image data.

Ln 204. It is not made clear in Figs 3A and 3G, what is the control sample/target, used to determine “relative mRNA expression. The reader is required to return to the Materials and Methods – where it is also not stated explicitly – and read Table 1 to discover this key detail. What is the evidence that the expression of GAPDH is invariant in these samples? Given evidence for cytotoxicity, expression of GAPDH may not be invariant, and thus render inaccurate all sqRT-PCR data!

Ln 209. Close examination of the WB data presented in Figs 3E and 3H, illustrates that these data is a composite of different blots derived from different experiments. For example the GAPDH control reported in Fig 3E is identical to the GAPDH control reported in Fig 3H. This is misleading. Scientifically, this is unacceptable.

Ln 227. For reproducibility, please specify how “weight loss of the implanted keloid tissue was measured”? Is this wet weight, or is this dry weight? Does this include encapsulating (i.e. host) tissue?

Ln 235. Sizing/scale bars are absent  from Fig 4a, and 4B.

Ln 229. What is the evidence for “less abundant collagen and micro-vessels in the derazantinib treatment groups”? For reproducibility, please identify how ‘abundance’ was measured? No detail is provided to identify “micro-vessels”. Please specify what method was used to identify, and verify, the presence of “micro-vessels”? (for example, anti-CD31, CD105, vWF, LYVE-1)

Ln 233. What is the evidence “the expression of a-SMA and PAI-1 was suppressed”? The evidence indicates that the detection of a-SMA and PAI-1 is less evident in treated tissue than the detection of a-SMA and PAI-1 in untreated tissue. There is no evidence for suppression!

Lns 245-248. The first paragraph of the discussion is superfluous. Remove!

Lns 249-255. The second paragraph repeats material discussed previously in the Introduction. Remove!

Ln 256. The first sentence of paragraph 3 is also a repeat of previous material.

Ln 258. This is where the discussion should start!

Ln 258. Please identify that the data being discussed here is in vitro data; please make clear that it is not in vivo data. Please also substantiate your experimental interpretations by cross-referring to the relevant data reported under Results.

Lns 264-271. I am not sure what value the sentences 264-271 add to the discussion. In this section, please discuss tour own data, and then cross-refer to published data from others.

You do this from line 279! Please introduce this approach earlier, preferably from the beginning of the Discussion section!

What is missing from this discussion is how might your findings be translated into clinical practice? How might your data be applied to the clinical management of keloids in humans? How might you persuade others that your findings are a viable approach to managing keloids? How does the use of derazantinib, as an intervention, compare with current therapeutic interventions? What are the next steps in developing these findings and capturing the value of your observations?

Comments on the Quality of English Language

The authors are not native speakers of English; the grammar is poor. The manuscript requires sub-editorial intervention.

Reviewer 3 Report

Comments and Suggestions for Authors

Introduction:

The introduction is far too short and many interesting details about derazantinib are missing, e.g. studies with patient numbers, mode of application (systemic, local, intravenous, injection, given amounts etc.).

Also, studies on FGF and its function would improve the reader's understanding.

Materials Method:

Number of patients for all experiments except immunofluorescence are missing . Here, you stated 8 Patients, but in the statistic you stated that all experiments were performed three times. (Three patients?, Every Patients 3times? Patients pooled) Please clarify!

More details for the mouse model! (n=?), age, race, induction of keloids, Weight loss analysis, Mas-162 son staining, and immunohistochemistry

Results:

3.1. FGFR1 demonstrated higher expression in keloid tissue

Figure 1 and the other figures needsto be fundamentally revised. The fluorescence in the photos is difficult to see and the size and labeling of the graphs are too small. Also,  the scale bars are often too tiny.

How do performed quantification of fluorescence signals? Relative fluorescence intensities to what? Per DAPI positive cell, per sight vs control. Please clarify in the method section!

3.2. Derazantinib reduced viability and promoted apoptosis of KFs

When exactly was the CKK measurement performed? The method section states a 48 h incubation and viability measurements after 24, 48, 72 and 96 h. The graph Fig. 1 B shows 1, 3, 5 and 7 days. This is very confusing and does not fit. Was the CKK8 measurement done after or during the 48 incubation with derazantinib?

Nevertheless, both 2.5 µM and 5 µM completely wiped of viable KFs!

In my opinion, it is not possible to state an apoptosis after a 48 incubation. Here, you should have done earlier staining 5-12 h after addition of derazantinib. Furthermore, Figure 1 D shows a lot of PI+ cells, therefore, dead cells without functioning membrane. To my knowledge a loss of Calcein AM-PI signal combined with anincrease of PI signal indicate necrosis.

By the way, I am wondering that the PCR experiments were performed after 48 h derazantinib incubation, whereas protein production of SMA and collagen after a 2 h incubation. Measureable changes in protein production should be later. Are you sure that the treatment did not affect the cells prior the western blot for example, by detachment of cells, possible undifferentiated (in proliferation)? This could also explain the loss of band signal, in particular a-SMA.

3.5. Derazantinib inhibited collagen production and disrupted angiogenesis of keloid tissue in an 223 athymic nude mouse model

Please clarify the weight loss measurement. It is no clear how you calculated the weight loss (%) of the control. Did you weight the implanted keloids tissues prior implantation? How many mice? In figure 4A I can see 4 samples, thus n=4?

Discussion

Please remove: Authors should discuss the results and how they can be interpreted from the per-245 spective of previous studies and of the working hypotheses. The findings and their impli-246 cations should be discussed in the broadest context possible. Future research directions 247 may also be highlighted. cations should be discussed in the broadest context possible. Future research directions

You stated: “In this study, we found higher expression of FGFR1 in keloid. To explore the 272 potential mechanism of derazantinib for keloid treatment, we validated the effects of de-273 razantinib on the TGF-β/SMAD signaling pathway, ERK, and AKT signaling pathway. 274 Our study showed that the expression and phosphorylation of SMAD were not signifi-275 cantly regulated by derazantinib while demonstrating apparent suppression of PI3K, p-276 AKT and p-ERK. Hence, we speculate that PI3K/AKT and ERK signaling pathway could 277 be responsible for the antiproliferative mechanism of derazantinib.”

I think you discussed the wrong paper, because for this paper you did not the experiments

Major points:

In general, derazantinib could work as keloid therapy, however, the shown experiments only allow the statement that the drug in the selected concentration (e.g. 2.5 µM) may have only a non-specific cell toxic effect, which of course inhibits proliferation, reduces cell number by cell death and proliferation inhibition, which all would explain pretty much all results. This could also obtained by e.g. subtoxic concentration of hxdrogen peroxide. Dead or dying and damaged cells do not proliferate and migrate anymore and the gene/protein expression of stressed cells are also significantly different. Also, it seems that the effect could be dependent on proliferation state or cell densitiy.

Why did you not perfomed experiments with normal fibroblasts? Here, you could prove a possible specifity of derazantinib.

However, it is desirable to have reliable evidence of specific apoptosis induction in KFs (proliferating, but also differentiated) by treatment, preferably at an early stage, which is a basic requirement for an effective keloid therapeutic agent. However, the presented results could also indicate necrosis and detachment of the cells, so that a definite statement is not possible. This would require many more experiments with controls, so unfortunately I cannot make a recommendation for publication.

Reviewer 4 Report

Comments and Suggestions for Authors

The authors study the keloids.  Studies have shown that fibroblast growth factor receptor- 12 1(FGFR1) was enhanced in pathological fibrous proliferation disease, such as cirrhosis and idiopathic pulmonary fibrosis (IPF), suggesting that FGFR1 pathway has potential for the keloid treatment, the authors propose Derazantinib  a selective FGFR inhibitor with the ability of antiproliferative activity in vitro and  in vivo models. The authors found that derazantinib inhibited the proliferation, migration, invasion and collagen production of  keloids in vitro. The transcription and expression of plasminogen activator inhibitor-1(PAI-1), which is  closely related to collagen deposition and tissue fibrosis, was significantly inhibited. Also, derazan tinib can inhibit the expression of FGFR1 and PAI-1 and reduce the weight of implanted keloid from 23 xenograft mice model. The authors conclude that that derazantinib may be the potent therapy for keloid via FGFR signaling.

The manuscript is well done and I have very few suggestios:

a) Please dedicate a space in the introduction to wound healing (years 2020-2022)

b) In the introduction write also a part dedicate to the therapies of keloids (years 2020-2022)

c) Please, always in the introduction, a space dedicated  to mast cells and their role in this pathology is essential (years 2020-2022)

d) Pleate check the acronyms and their use in the manuscript

Comments on the Quality of English Language

Moderate editing of English language required

Reviewer 5 Report

Comments and Suggestions for Authors

In this study, the effects of derazantinib on keloid fibroblasts were studied with promising results. The major issue, however, is the lack of use of normal fibroblasts. To demonstrate any specific effect of derazantinib on keloid fibroblasts, normal fibroblasts and tissue should be included in the various tests.

Abbreviations (KFs, DZB, BMT) should be written in full the first time.

Materials and Methods

Please provide more details for secondary antibodies (2.1), seeding density (2.5), type/company for Boyden chamber (2.6), age, weight, number of animals, implantation technique, method for weight loss (weighing before implantation?) (2.11). Please check and include a more extensive protocol (2.1). How can you remove the epidermis by using scissors? This will also remove a large layer of the papillary dermis. The used enzymes should be stated in U/ml instead of %.

Three independent experiments: does this mean 3 x 4 mice? Or is it the number of replicates in each experiment? When using Student’s T test, a check for normal distribution should be done.

Why not use Annexin V for apoptosis?

Results

Fig 1A: how was the signal quantified? What was the number of samples/donors? Fig 1B: what is the effect on normal fibroblasts? Why use 2.5 µM in most of the consecutive tests and not 0.63 or 1.25 µM? Please include images with a higher resolution. Although the signal of Calcein AM is lower, the number of cells positive for Calcein AM seems similar, please comment.

Line 212 typo 4D should be 3H.

Fig 3A & 3H: graphs and text are very small, please increase size. In fig 3A, how was relative expression calculated (to what)? The CTRL is in all case just below 1. Please include more experimental details in all figure captions: method, culture/treatment time etc.

3.5 How do you explain the reduced expression of FGFR1 in vivo (4C) while there was no effect on FGFR1 in vitro (3A)? line 231.

Fig 4. How specific was removal of implanted tissue from the mice? There was only one mice + 4 biopsies per treatment? What was the weight before implantation, or how sensitive/accurate is weighing these samples (in the range of a few mg)?

Please remove lines 245-247.

Comments on the Quality of English Language

Spelling and grammar need to be checked (minor). 

Author Response

Please see the attachent.

Round 2

Reviewer 1 Report

Comments and Suggestions for Authors

Some of the original comments have not been addressed by the authors:

1) The study of the inhibitor effects should focus primarily on the phosphorylation status of FGFR1, since this is a kinase inhibitor. Otherwise the authors should explain how a kinase inhibitor leads to lower FGFR1 expression levels.

2) Only calcein AM is quantitatively assessed in Figure 1D. The authors should also count cells positive for PI and show the relevant histogram.

Comments on the Quality of English Language

No comments.

Author Response

On behalf of all the contributing authors, I would like to express our sincere appreciations of your professional review work on our article. These comments are all valuable and helpful for improving our article. As you are concerned, there are several problems that need to be addressed. According to your nice suggestions, we have made extensive corrections to our previous draft, the detailed corrections are listed below.

Reviewer 2 Report

Comments and Suggestions for Authors

The premise of this manuscript is based on cumulative evidence from many sources that FGF is a ‘Master Controller’ of keloid development and growth. The authors “hypothesized that derazantinib (a FGFR1‒3 kinase inhibitor, which also recognises colony stimulating factor-1‒receptor (CSF1R) and vascular endothelial growth factor receptor‒2 (VEGFR2)) could attenuate keloid formation by directly targeting KFs. To test this hypothesis, the authors detected the expression of FGFR1 in keloid tissue and normal skin, …tested derazantinib bioactivity on keloid fibroblasts, and… evaluated derazantinib in an athymic nude mouse model.”

In my opinion this current version still lacks sufficient detail to allow others to reproduce the reported experiments. Too many protocols are not reported (for example: serum-starvation, SDS-PAGE, Masson staining, HE staining). Figure legends remain inadequate; most do not report used annotations, protocols, or statistical tests.

Ln 37. Please replace “recurrence rate” (incorrect use) with ‘incidence of recurrence.

Ln 41. What is the evidence “Single-cell RNA sequencing reveals a significant increase in fibroblasts, endothelial cells, mural cells, and mast cells in keloid tissue”? (sic) A reference is required.

Actually, I now see that this evidence is cited after the following sentence; however, does not substantiate that sentence! Please revise.

Ln 51. This reader is confused. What is meant by “…the clinical therapeutic effects of keloids”? (sic) I was unaware that keloids possess any “clinical” or “therapeutic effects”. I interpret that you mean “…the therapeutic effects…” of clinical interventions on “keloids”.

Ln 53. Please elaborate. What is “conservative treatment”? (sic)

Ln 97. It is more usual for tissue to be “fixed in 4% paraformaldehyde” prior to being “sectioned” (sic) Would the average, non-informed reader understand the meaning of “sectioned”?

Ln 106. What is meant by “samples were obtained through operations”? (sic) For the purpose of reproducibility, please identify “operations”?

Ln 117. Please clarify “independent cell samples”? Does this mean assay samples were sourced from: independent donors? Independent tissues? Independent culture vessels? Independent cell passages” In other words, please clarify whether samples were technical replicates, assay replicates, biological (unrelated donor) replicates?

Ln Although stored as “stock concentration of 10 mM… at −20°C”, please specify what measures taken to ensure DMSO was anhydrous?

Ln 136. What is the meaning of “KFs were planted seeded…”? I suggest this is an editing widow?

Ln 137. Please identify what measures were taken to remove displaced cells after “cell monolayer was scraped with a 200μl pipette tip”? This is an essential step to minimise displaced cells reattaching and contributing to ‘scrape-wound healing’ and thus invalidate assay measures.

Ln 139. The statement that “Photographs were taken after the scratching at the indicated time points” (sic) does not report how “Cell Migration… of KFs” was measured! Please identify how the cell migration assay was quantitated?

Ln 142. Please identify, what is the origin of “serum-starved KFs”?

The starvation protocol is important to the metabolic state and viability of the explant-derived cells. Being primary cells, it is very unlikely they become synchronized. Rather, “serum-starved KFs” may be metabolically stressed, refractory to growth factor stimulation, exhibit dysregulated secondary messenger pathways, and potentially are pro-apoptotic. This ensures demonstrating reproducibility is challenging. How were serum-starved cells validated to be representative of the parent cell population?

Ln 145. For the benefit of reproducibility, please specify the diluent for “0.1% crystal violet”? (sic)

Ln 147. It is not made clear what parameters were measured in the “Cell Viability Staining” protocol.

Ln 183. Please specify the formulation for “Radioimmunoprecipitation assay (RIPA) buffer”? A reference will suffice.

Ln 185. As currently written, the author’s assume all readers know the protocol for “SDS-PAGE” and for “transfer… onto a polyvinylidene fluoride (PVDF) membrane”! (sic) Please assist less-than-expert readers to identify these common protocols by citing a source reference?

Ln 186. Please specify what is the pronoun “it” referring to, in “It was subsequently treated…” 

Please identify “the specific primary antibodies”?

Ln 187. For the benefit of reproducibility, please specify “horseradish peroxidase-conjugated secondary antibody”? 

Ln 193. Delete “Keloid tissue… was implanted”. This is repeated in line 198.

Ln 204. While protocols are reported for “Immunofluorescence assay of KFs”, protocols are not reported for “Masson staining, and immunohistochemistry”. For the benefit of reproducibility, please report ALL protocols.

Ln 219. What do the authors mean by “almost completely wiped off viable KFs”? (sic)

Ln 228. It is not obvious, not is it reported; what is the parameter “Merge” (sic) in Figure 1(A) and 1(C)?

It is not reported in the Figure Legend (Lns 229-234) what the Bar Charts to the right of image data reported in Figure 1(A), 1(C), and 1(D), represent? How these data were obtained is also not reported. 

Ln 240. Why is the cell distribution on Transwell™ membranes reported in Figure 2C non-uniform? These images reveal high cell density evident on the left side, that is not replicated on the right side. These data suggest to this reviewer that these assay data are biased. For reasons that are not clear, these data are not robust evidence for cell invasion.

Ln 247. What is the evidence that “collagen I, αSMA, and PAI-1”, are “fibrotic factors”? (sic) 

Ln 249. Would the average reader understand what are “marker molecules”? (sic)

Ln 256. The data reported in Figure 3E does not report “protein production”; rather, these data report relative immunoreactive masses of target proteins aSMA, collagen I, PAI-1 and FGFR1. No assay was performed, nor are any data reported for “protein production” (aka biosynthesis). It cannot be concluded that changes in protein mass correlate with changes protein function. Please revise.

Ln 258. The data reported in Figure 3H does not report “protein production”; rather, these data report relative immunoreactive masses of ERK, p-ERK, AKT, p-AKT and PI3K. These data do not report “expression of ERK, p-ERK, AKT, p-AKT and PI3K” as is claimed in the Figure Legend text. Please revise.

Ln 276. The claim that “No significant effect (of treatment with derazantinib) was found on TGF-β” is not supported by the evidence reported in Figure 3H. The data presented claimed to represent TGF-β is a doublet species in the control lane, but is predominantly a singlet species in the “2.5 µM” lane. While not specified, the authors presume the reader understands that data annotated as “2.5 µM”, are samples from cells exposed to 2.5 µM derazantinib for 48 hours. There are no experimental methods reported in Figure Legends; thus the reader must reread the Materials and Methods to understand the figure. This is lazy, inadequate communication, and unacceptable.

Ln 284. This reader is unable to locate any description under Materials and Methods, (or elsewhere), for “Masson staining and HE staining” (sic).

Ln 285. Ln 292. No evidence is reported in Figure 4B to substantiate the claim that “staining showed less abundant collagen and micro-vessels in the derazantinib treatment groups”. (sic) I interpret that this claim is based on visual identification of structures that ‘look like’ “collagen and micro-vessels’; however, alternative and corroborating evidence is not reported.

Ln 302. What is meant by “…unsatisfactory clinical treatment”? 

Ln 305. This is reader is confused. I do not understand how both “fibroblasts and fibrosis-related genes” may be “amplified”? (sic)

Ln 314. Please define IPF within the main text. It is mentioned in the Abstract, but not the main text.

Ln 317. I disagree with the interpretation “excessive PAI-1 contributes to excessive accumulation of collagen and other ECM protein around the wound, thus preserves scarring.” Other evidence indicates elevated PAI-1 have pleiotropic effects; in some situations PAI-1 is profibrotic, in others PAI-1 inhibits fibrosis.  I recommend a more nuanced perspective is necessary.

Ln 327. What is the evidence “PAI-1… mediat(es) the inhibition of derazantinib in keloid”? (sic)

Ln 336 What is the evidence “…systemic (administration…) of derazantinib… is safe”? 

Formatting of some references is incorrect – spaces between parameters are absent.

Ln 361. Reference 3 is incomplete (missing page numbers)

Ln 377. Reference 10 is incomplete (missing page digital object identifier)

Ln 409. Reference 22 is incomplete (missing page numbers)

Comments on the Quality of English Language

It is evident the authors native language is not English.

The text requires sub-editorial intervention to correct basic sentence structure and grammar. 

Author Response

On behalf of all the contributing authors, I would like to express our sincere appreciations of your professional review work on our article. These comments are all valuable and helpful for improving our article. As you are concerned, this current version still lacks sufficient detail to allow others to reproduce the reported experiments. According to your nice suggestions, we have made corrections to our previous draft, the detailed corrections are listed below.

Reviewer 3 Report

Comments and Suggestions for Authors

Some of the illustrations are not beautiful. Especially the graphs are often too small and their labels hardly readable. In addition, the number of experiments and whether they are mean values with standard deviations are missing in the legends.

I still have problems with the stastistics: How can 8 patients or 12 mice be tested in only 3 independent experiments?  Unfortunately, I could not find the improvements regarding this problem.

How the relative fluorescence intensities were determined is still not in the manuscript or legends.

The number and data of patients for obtaining normal skin is missing as well as the number of CKK experiments with normal fibroblasts.

An important point was also not discussed: It seems that normal fibroblasts are not as sensitive to DZB as keloid fibroblasts, which is a good thing!

The DZB-induced toxicity is still there and whether this is due to apoptosis should be discussed more. Also that the DZB-induced toxicity and cell damage at the chosen high DZB concentrations could also have a cause in the observed decreased migration, collagen, SMA and PAI expressions needs to be discussed a little bit more.

Here, I recommend a manuscript section “Limitation”

The original images of all blots done used for the calculation of protein expression  are missing!

Author Response

(The authors gave the same response as above.)

Reviewer 5 Report

Comments and Suggestions for Authors

Thank you for the answers.

Response 1. I assume that a concentration of 2.5 μM did not provoke apoptosis and death of the normal fibroblasts (not keloid fibroblasts)? See also point 6. Did a higher concentration affect normal FB? I would advise to use normal FB in all in vitro experiments. migration and collagen production might be affected. The number of cells positive for Calcein AM can be included in paper.

Point 4. A check for normal distribution was not performed. It should be mentioned in the figure caption that n=12 animals and mention that n=3 for keloid tissue.

Point 8. Shouldn’t the CTRL be equal to 1 in all cases instead of just below 1? Stable expression of GAPDH in different conditions should be demonstrated. It is better to use 3 different reference genes. See for example this paper: Reference genes in real-time PCR. J Appl Genet. 2013; 54(4): 391–406. PMID: 24078518 by Kozera and Rapacz. They stated that “experimental confirmation of the stability of candidate genes is now a standard requirement.”

In the revision:

2.4 line 130. I assume this is in µmol/ml?

2.9 line 175. Typo GAPDH

2.11 What was the source of the keloids, what was the procedure (consent, location, age, etc.). Since keloids from 3 patients were used, I wonder if the results should be averaged per keloid donor, before doing statistics.

In several figures, the font sizes (and graphs) are too small.  Figure 1 & 3, there is actin (red) shown?

Figure 4A: these are the results of 1 keloid in 4 animals? For clarity, please add the animal number above the samples.

The discussion is rather limited.

Author Response

(The authors gave the same response as above.)

Round 3

Reviewer 1 Report

Comments and Suggestions for Authors

Despite the authors' reply, I don't see the relevant revisions in the current manuscript version. In particular:

Regarding Point 1 (The study of the inhibitor effects should focus primarily on the phosphorylation status of FGFR1, since this is a kinase inhibitor. Otherwise the authors should explain how a kinase inhibitor leads to lower FGFR1 expression levels) the authors' reply was "Thank you for your advice. The limitation is the lack of detection of FGFR1 and other kinases phosphorylation status. We will complete the experiments in further research. Lower FGFR1 expression may be due to cell apoptosis or necrosis." However, this limitation, as well as, the possible cause for lower FGFR1 expression should be incorporated into the Discussion of the manuscript.

Regarding Point 2 (Only calcein AM is quantitatively assessed in Figure 1D. The authors should also count cells positive for PI and show the relevant histogram) the authors' response was "We have made corrections". However, no histogram for PI-positive cells has been added to Figure 1D.

Comments on the Quality of English Language

No comments

Reviewer 2 Report

Comments and Suggestions for Authors

This is the second revision of this manuscript that I have reviewed. 

The authors have read, digested, and responded to all of my previous suggestions and recommendations. They have amended the text which now reads clearly. 

I am happy to recommend this version is suitable for acceptance and publication.

Comments on the Quality of English Language

Adequate

Author Response

On behalf of all the contributing authors, I would like to express our sincere appreciations of your professional review work on our article. These comments are all valuable and helpful for improving our article. Thank you.

Reviewer 3 Report

Comments and Suggestions for Authors

Point 1: Some of the illustrations are not beautiful. Especially the graphs are often too small and their labels hardly readable. In addition, the number of experiments and whether they are mean values with standard deviations are missing in the legends.

Response 1: We have made corrections for the illustrations.

è  I can not observe any improvement of the illustrations, , the labels still too small and the number of experiments and kind of values (mean SD median ND?  still missing.

Point 2: I still have problems with the stastistics: How can 8 patients or 12 mice be tested in only 3 independent experiments? Unfortunately, I could not find the improvements regarding this problem.

Response 2: The assay was repeated with three cell samples from independent donors. In an athymic nude mouse model(n=12), keloid tissue from each patient(n=3) were transplanted subcutaneously into 4 mice respectively.

è  Thus, n=3 with 4 replicates

Point 3: How the relative fluorescence intensities were determined is still not in the manuscript or legends.

Response 3: Image-Pro Plus was used for fluorescence intensity analysis.

à Done!

Point 4: The number and data of patients for obtaining normal skin is missing as well as the number of CKK experiments with normal fibroblasts.

Response 4: Normal skin was harvested from six female patients (aged 27 to 42 years) who were performed with breast reduction operations. The CCK8 assay with normal fibroblasts was repeated with three cell samples from independent donors.

à Good to know, please state it also in the method section.

Point 5: An important point was also not discussed: It seems that normal fibroblasts are not as sensitive to DZB as keloid fibroblasts, which is a good thing!

Response 5: Although, derazantinib at 2.5 μmol/L showed different inhibitory effects on KFs and Fbs, it inhibited the proliferation of Fbs on higher concentrations. Further research will be performed to explore the effect of DZB on Fbs. 2

àDone!

Point 6: The DZB-induced toxicity is still there and whether this is due to apoptosis should be discussed more. Also that the DZB-induced toxicity and cell damage at the chosen high DZB concentrations could also have a cause in the observed decreased migration, collagen, SMA and PAI expressions needs to be discussed a little bit more.

Response 6: Despite the pleiotropic effect on KFs, toxic effects of the drug on the cells cannot be completely ex-cluded. This is the limitation of the drug.

àdone

Point 7: The DZB-induced toxicity is still there and whether this is due to apoptosis should be discussed more. Also that the DZB-induced toxicity and cell damage at the chosen high DZB concentrations could also have a cause in the observed decreased migration, collagen, SMA and PAI expressions needs to be discussed a little bit more. Here, I recommend a manuscript section “Limitation”. The original images of all blots done used for the calculation of protein expression are missing!

Response 7: Thank you for your advice. We have made corrections.

àDone!

àThe original images of all blots are still missing

Reviewer 5 Report

Comments and Suggestions for Authors

A major issue is the lack of normal tissue in the animal tests. To demonstrate any specific effect of derazantinib on keloid fibroblasts, normal tissue should be included. I realize this is not easily solved but it is scientifically crucial.

Response 1 should be mentioned in the text as well (cell death of normal FB at 5 µM).

Corrections for point 2 were not made.

Point 3.  Limiting the reference gene to only GAPDH may result in wrong interpretation of the results. It must be demonstrated that GAPDH expression is not affected by DZB for example.

Some figures/fonts are still quite small.

I don't see the corrections for point 7.

Discussion is still limited.

Round 4

Reviewer 1 Report

Comments and Suggestions for Authors

The authors have addressed all comments.

Comments on the Quality of English Language

No comments.

Reviewer 5 Report

Comments and Suggestions for Authors

I have no further comments